

# Enhanced specialized metabolite, trichome density, and biosynthetic gene expression in *Stevia rebaudiana* (Bertoni) Bertoni plants inoculated with endophytic bacteria *Enterobacter hormaechei*

Dumas G. Oviedo-Pereira[1], Melina López-Meyer[2], Silvia Evangelista-Lozano[1], Luis G. Sarmiento-López[2], Gabriela Sepúlveda-Jiménez[1] and Mario Rodríguez-Monroy[1]

[1] Biotecnología, Instituto Politécnico Nacional Centro de Desarrollo de Productos Bióticos, Yautepec, Morelos, México

[2] Departamento de Biotecnología Agrícola, Instituto Politécnico Nacional. Centro Interdisciplinario de Investigación Para el Desarrollo Integral Regional (CIIDIR), Guasave, Sinaloa, México

Corresponding author
Mario Rodríguez-Monroy,
mrmonroy@ipn.mx

## ABSTRACT

*Stevia rebaudiana* (Bertoni) Bertoni is a plant of economic interest in the food and pharmaceutical industries due its steviol glycosides (SG), which are rich in metabolites that are 300 times sweeter than sucrose. In addition, *S. rebaudiana* plants contain phenolic compounds and flavonoids with antioxidant activity. Endophytic bacteria promote the growth and development and modulate the metabolism of the host plant. However, little is known regarding the role of endophytic bacteria in the growth; synthesis of SG, flavonoids and phenolic compounds; and the relationship between trichome development and specialized metabolites in *S. rebaudiana*, which was the subject of this study. The 12 bacteria tested did not increase the growth of *S. rebaudiana* plants; however, the content of SG increased with inoculation with the bacteria *Enterobacter hormaechei* H2A3 and *E. hormaechei* H5A2. The SG content in leaves paralleled an increase in the density of glandular, short, and large trichome. The image analysis of *S. rebaudiana* leaves showed the presence of SG, phenolic compounds, and flavonoids principally in glandular and short trichomes. The increase in the transcript levels of the *KO*, *KAH*, *UGT74G1*, and *UGT76G1* genes was related to the SG concentration in plants of *S. rebaudiana* inoculated with *E. hormaechei* H2A3 and *E. hormaechei* H5A2. In conclusion, inoculation with the stimulating endophytes *E. hormaechei* H2A3 and *E. hormaechei* H5A2 increased SG synthesis, flavonoid content and flavonoid accumulation in the trichomes of *S. rebaudiana* plants.

## INTRODUCTION

Plant–microbiome interactions are beneficial because they enhance the acquisition of mineral nutrition and provide protection against abiotic and biotic stresses in plants (*Asaf et al., 2017*; *Wang et al., 2015*). The study of these interactions has great potential for application in biotechnology and agriculture because the use of microorganisms improves the plant growth and development of food crops (*Lodewyckx et al., 2002*; *Rosenblueth & Martínez-Romero, 2006*).

Endophytic bacteria are an important group of microorganisms that are found in different plant tissues, such as in the roots (rhizosphere), leaves (phylloplane), stems (laimosphere and caulosphere), fruits (carposphere), seeds (spermosphere) and flowers (anthosphere) (*Brader et al., 2017*). In this relationship, plants and endophytic bacteria form a unique interaction with the ability to provide alternative sources of active metabolites such as enzymes, biofunctional chemicals, phytohormones and nutrients and to facilitate the distribution and production of secondary metabolites (*Hardoim et al., 2015*; *Santoyo et al., 2016*). The host plant provides a protective environment for the bacteria, in which the microorganism can grow and reproduce, but with no adverse effects that negatively affect plant growth and health (*Shahzad et al., 2018*). Bacteria also enhance the accumulation of secondary metabolites and modulate the accumulation profile and the expression patterns of several biosynthetic pathways in many plant species (*Tiwari et al., 2013*; *Tiwari et al., 2010*; *Yang et al., 2019*; *Zhou et al., 2016*). For example, isolated bacteria from *Lycoris radiata (L'Hér.)* Herb promote Amaryllidaceae alkaloid accumulation in the host plant (*Liu et al., 2020*), and *Pseudomonas fluorescens* induces sesquiterpenoid accumulation in *Atractylodes macrocephala* Koidz plants (*Yang et al., 2019*).

Trichomes are epidermal structures where various secondary metabolites are synthesized and accumulated and are associated with the chemical defense of the plant (*Li et al., 2020*; *Werker, 2000*). Trichomes are classified according to their morphology into glandular and nonglandular groups. In particular, glandular trichomes play an important role in the deposition of many secondary metabolites, such as alkaloids, polyketides, phenylpropanoids, phenolic compounds and terpenoids (*Tian et al., 2017*).

*Stevia rebaudiana* (Bertoni) Bertoni is a perennial shrub species of the Asteraceae family and is an economically important crop due its ability to accumulate specialized metabolites called steviol glycosides (SG), including isosteviol, stevioside, rebaudiosides (A, B, C, D, E and F), steviolbioside and dulcoside A, which are used as low-calorie sweeteners (*Sarmiento-López et al., 2020*; *Rajasekaran et al., 2008*). The sweet taste of *S. rebaudiana* leaves depends on the contents of stevioside and rebaudioside A, which are approximately 250–300 times as sweet as sucrose (*Geuns, 2003*). Due to the high content of sweet glycosides, *S. rebaudiana* is considered a valuable source of natural sweeteners for the growing food market (*Goyal & Goyal, 2010*). In addition, the leaves of *S. rebaudiana* contain phenolic compounds, which are a family of antioxidant metabolites, including stilbenes, flavonoids and phenolic acids (*Lemus-Mondaca et al., 2012*).

*Brandle & Telmer (2007)* proposed that SG biosynthesis begins with geranylgeranyl-di-phosphate (GGDP) synthesis through the methyl-erythrol-4-phosphate (MEP) route.

GGDP is transformed to kaurene by two cyclization steps carried out by terpene cyclases and later converted to steviol by four additional enzyme actions: (EC 5.5.1.13) copalyl diphosphate synthase (*CDPS*), (EC 4.2.3.19) kaurene synthase (*KS*), (EC 1.14.14.86) kaurene oxidase (*KO*), and kaurenoic acid hydroxylase (*KAH*) (*Kim, Sawa & Shibata, 1996*). Different SG are formed by steviol glycosylation by specific glucosyltransferases; the enzyme (EC 2.4.1.17) *UGT74G1* is involved in the conversion of steviolbioside to stevioside, while the enzyme (EC 2.4.1.17) *UGT76G1* is involved in the conversion of stevioside to rebaudioside A (*Shibata et al., 1991*; *Shibata et al., 1995*). Some studies have been carried out in *S. rebaudiana* to evaluate the effect of plant growth-promoting rhizobacteria (PGPR) and mycorrhizal fungi on growth, secondary metabolite accumulation, and the expression of biosynthetic genes. *Mamta Rahi et al. (2010)* and *Vafadar, Amooaghaie & Otroshy (2014)* reported that inoculation with different PGPR improved plant growth, photosynthetic parameters, and the accumulation of stevioside and rebaudioside A. Likewise, *Sarmiento-López et al. (2020)* reported that arbuscular mycorrhizal (AM) symbiosis with *Rhizophagus irregularis* improves growth and photosynthetic activity. Additionally, they reported the upregulation of the biosynthetic genes *KO, UGT74G1* and *UGT76G1*. Furthermore, it has been proposed that the synthesis and accumulation of SG take place in trichomes (*Bondarev et al., 2010*). Recently, *Sarmiento-López et al. (2021)* reported that AM symbiosis with *R. irregularis* induced a significant increase in the accumulation of phenolic compounds, related to the high number of trichomes, and reported that these metabolites were localized specifically in the secretory cavity of glandular trichomes.

Endophytic bacteria are microorganisms that can live inside plant tissues, providing advantages over other rhizospheric microorganisms. However, little is known regarding their role in secondary metabolism, plant growth, and the relationship between trichome development and specialized metabolites (SG and phenolic compounds) in *S. rebaudiana*. In this study, we hypothesize that (1) inoculation with endophytic bacteria promotes the accumulation of specialized metabolites and the expression of their biosynthetic genes in *S. rebaudiana* and (2) endophytic bacteria induce the development of trichomes in relation to the accumulation of specialized metabolites. Thus, the objective of this work was to evaluate the accumulation of specialized metabolites and the expression of their biosynthetic genes in *S. rebaudiana* and the development of trichomes in response to inoculation with endophytic bacteria of *S. rebaudiana* plants.

## MATERIALS AND METHODS

### Growth of *Stevia rebaudiana* plants

*S. rebaudiana* plants were grown under greenhouse conditions at Centro de Desarrollo de Productos Bióticos (CeProBi-IPN) in Morelos, México, according to the methodology described by *Sarmiento-López et al. (2021)*. Briefly, to obtain rooted plants, one apical shoot (three to five cm long) was planted for each 1 $dm^3$ pot containing a mixture of 60:20:20 (w:w:w) sterilized turf, perlite, and vermiculite with an initial pH of 5.6 ± 0.5 and a porosity of 85%. This substrate was sterilized at 121 °C and 15 psi for 2 h. The plants growing in this substrate showed root formation at 15 days.

**Table 1** Identification of endophytic bacteria isolated from *Stevia rebaudiana* plants.

| Tissue | Isolate | Most similar strain NCBI database | Access code | Identity |
|---|---|---|---|---|
| Leaf | H2A3 | *Enterobacter hormaechei strain* C15 | CP042488.1 | 99.7% |
| Leaf | H5A2 | *Enterobacter hormaechei strain* ER48 | MT124573.1 | *100%* |
| Leaf | H7A1 | *Enterobacter bacterium strain E* Mt 5 | EU863187.1 | *90%* |
| Stem | T1A2 | *Enterobacter xiangfangensis strain* KV7 | MH200641.1 | 99.7% |
| Stem | T3A3 | *Enterobacter xiangfangensis strain* SitB416 | KY880912.1 | 99.8% |
| Stem | T5P1 | *Enterobacter xiangfangensis strain* KV7 | MH200641.1 | 99.7% |
| Root | R2A2 | *Enterobacter xiangfangensis strain* KV7 | MH200641.1 | 99.7% |
| Root | R3A1 | *Enterobacter cloacae strain* MB0-11 | MH041191.1 | *100%* |
| Root | R5P1 | *Enterobacter hormaechei strain* EGYMCRVIM | CP052870.1 | 99.3% |
| Root | R6A1 | *Enterobacter hormaechei strain* C45 | CP04255.1 | 99.7% |
| Root | R6P1 | *Bacillus safensis strain* F14 | MH065717.1 | *100%* |
| Root | R7A2 | *Enterobacter xiangfangensis strain* KV7 | MH200641.1 | *99.7%* |

## Endophytic bacterial culture

The endophytic bacteria were isolated from different tissues of *S. rebaudiana* plants: leaf, stem, and roots (Table 1). The tissues were rinsed with sterile water and surface sterilized using 70% ethanol (10 min) and 2% sodium hypochlorite (20 min). The fragments of each tissue were seeded in Petri dishes with LB medium and agar (Sigma–Aldrich, St. Louis, Missouri, USA); the Petri dishes were incubated at $25 \pm 1\,°C$ for 24 h. Axenic cultures were obtained and cryopreserved in glycerol at 20% (v/v) at $-80\,°C$. The 16S rDNA sequences of the isolates were compared to the GenBank database using BlastN and a phylogenetic analysis using the MEGA 6 program according to *Montes-Salazar, Maldonado-Mendoza & Rodríguez-Monroy (2018)*. The bacterial inoculum was grown in 250 $cm^3$ flasks with a volume of 100 $cm^3$ of liquid LB medium and incubated on a rotary shaker (Infors HT, Minitron, Switzerland) at 200 rpm for 48 h at 25 °C.

## Inoculation of *S. rebaudiana* with endophytic bacteria

Fifteen-day-old *S. rebaudiana* plants with five-cm-long roots and two leaves were used. Plants were disinfected by using 70% ethanol for 1 min, followed by 2% sodium hypochlorite for 1 min, and subsequently rinsed three times with sterile distilled water for 2 min. The plants were planted in 1 $dm^3$ pots containing the same substrate mentioned above. One day after being planted (time 0 from the start of the experiment), the plants were inoculated at the root with five $cm^3$ of culture broth of each of the 12 isolates. The concentration was adjusted to 0.2 OD at 600 nm (approximately $1 \times 10^8$ cells $cm^{-3}$) (*Botta et al., 2013*). The plants were grown at 28 °C, with a photoperiod of 16 h light/8 h darkness. Ten plants per treatment were considered, and two independent experiments were carried out. The control was noninoculated plants. All plants were watered every other day with a 50% Steiner solution (*Rodriguez-García, 2015*). The pots were placed in the nursery in a random arrangement, and no pruning was performed during the evaluation time.

## Evaluation of plant growth

The *S. rebaudiana* plants inoculated with the endophytic bacteria were collected at 30 days postinoculation (dpi). With a Vernier caliper, the plant height was measured from the surface of the substrate to the apex of the plant, and root size was measured from the base of the stem to the root apex. The numbers of leaves and shoots were recorded, and roots were separated and dried in an oven (RiossA E-33, Monterrey, México) at 50 °C. The dry tissue was weighed on an analytical balance, and the dry weight (DW) was recorded. For the biochemical determinations, the collection of leaves of inoculated and noninoculated plants was carried out following the method previously reported by *Sarmiento-López et al. (2021)*.

## Determination of steviol glycoside (SG) concentration

In the leaves of inoculated and noninoculated plants, the SG concentration was determined according to the methodology reported by *Villamarín-Gallegos et al. (2020)*. Briefly, the leaves were dried in an oven (RiossA E-33) at 65 °C for 48 h. Dry tissue (0.1 g) was extracted with 1 cm$^3$ of methanol (JT Backer, Phillipsburg, USA) in 1.5 cm$^3$ microtubes, according to *Woelwer-Rieck et al. (2010)*. The microtubes were stirred for 3 min, allowed to stand for 24 h without stirring, and then centrifuged at 1,300× g at 4 °C for 10 min. The supernatant was recovered, placed in fresh microcentrifuge tubes, and stored at 4 °C until analysis by high-performance thin layer chromatography (HPTLC, CAMAG, Muttenz, Switzerland). Stevioside and rebaudioside A concentrations were expressed as mg g DW$^{-1}$. For each treatment, three plants were evaluated, and two independent experiments were performed.

## Determination of phenolic compound and flavonoid concentrations

Samples (0.1 g) of dry leaves from plants not inoculated and inoculated with endophytic bacteria were extracted with 1 cm$^3$ of 75% ethanol and centrifuged at 1,300× g at 4 °C for 10 min. The supernatant was recovered in 1.5 cm$^3$ microcentrifuge tubes and kept at 4 °C until processing.

The phenolic compounds were determined using the Folin-Ciocalteu reagent as described by *Bobo-García et al. (2014)*. The reaction was performed on a microplate incubated at room temperature in the dark for 2 h. The absorbance was measured at 760 nm on a spectrophotometer (Multiscan Go, Thermo Fisher Scientific, Massachusetts, USA) equipped with SkanIt Software version 1.00.40. Gallic acid (Sigma–Aldrich, St. Louis, Missouri, USA) was used as a standard, and the curve was constructed with serial dilutions (5, 10, 15, 20, 25 μg cm$^{-3}$) in distilled water. The standard curve had a correlation value $R^2 = 0.995$. The results were expressed as mg equivalents of gallic acid (GAE) g DW$^{-1}$.

The flavonoid concentration was determined according to *Villamarín-Gallegos et al. (2020)* and adapted from *Chang et al. (2002)*. The assay mix was performed on a microplate of 96 wells and incubated at room temperature in the dark for 30 min. Absorbance was monitored at 415 nm on a spectrophotometer (Multiscan Go, Thermo Fisher Scientific) equipped with SkanIt Software version 1.00.40. Serial dilutions (5, 10, 15, 20, 25 μg cm$^{-3}$) of quercetin (Sigma–Aldrich) in distilled water were used to construct the standard curve; the correlation value of the standard curve was $R^2 = 0.995$. The results were expressed as mg equivalents of quercetin (QE) g DW$^{-1}$.

## Trichome analysis by environmental scanning electron microscopy (ESEM) and confocal laser scanning microscopy

The trichome density of leaves was analyzed with an environmental scanning electron microscope (EVO LS10; Carl Zeiss, Oberkochen, Germany) according to the methodology reported by *Sarmiento-López et al. (2021)*. Fully developed leaves close to the apical meristem from plants that were noninoculated and those inoculated with the endophytic bacteria were collected. A leaf was placed in aluminum stubs with double-sided conductive carbon tape and observed under ESEM using a voltage of 15 kV. The gas pressure in the ESEM chamber was maintained at 20 Pa by introducing water vapor, and a secondary electron detector was utilized to obtain micrographs. The trichome density in 0.255 $cm^2$ (trichome leaf area$^{-1}$) and the type of trichomes (short, large and glandular) were determined by image analysis using ImageJ editing software 2.0 from micrographs obtained by ESEM.

The effect of the inoculation of plants with endophytic bacteria on specific metabolite accumulation was visualized with a confocal laser scanning microscope (Carl Zeiss, model LSM 800, Germany) according to the methodology reported by *Sarmiento-López et al. (2021)*. The maximum fluorescence of secondary metabolites (SG, phenolic compounds and flavonoids) was observed in the blue spectrum (435–485 nm), and chlorophylls in the red spectrum (630–685 nm) were detected according to the methodology of *Talamond, Verdeil & Conéjéro (2015)*. Micrographs were obtained using Zeiss Efficient Navigation (ZEN) 2.6 Blue edition.

## Expression analysis by qRT–PCR

The transcript accumulation levels of the genes for kaurene oxidase (*KO*), kaurene hydroxylase (*KAH*) and (UDP)-glycosyltransferases (*UGT74G1* and *UGT76G1*) were evaluated in leaves of noninoculated plants (control) and leaves of plants inoculated with the selected endophytic bacteria *E. hormaechei* H2A3, *E. hormaechei* H5A2, and *E. xiangfangensis* R7A2. Expression of each gene was normalized against the expression levels of the housekeeping gene *GAPDH*. Frozen leaf samples (0.5 g DW) were ground to a fine powder with liquid nitrogen. Total RNA was obtained using TRIzol reagent (Invitrogen, Carlsbad, CA) following the manufacturer's protocol. First-strand cDNA synthesis was performed as previously reported by *Sarmiento-López et al. (2020)*.

The primers corresponding to the *KO* gene were SrKOF 5′-TCTTCACAGTCTCGGTGG TG-3′ and SrKOR 5′-GGTGGTGTCGGTTTATCCTG-3′; the primers corresponding to the *KAH* gene were SrKAHF 5′-CCTATAGAGAGGCCCTTGTGG-3′ and SrKAHR 5′-TAGCCTCGTCCCTTTGTGTC-3′; the primers corresponding to the glycosyl-transferase *UGT74G1* gene were SrUGT74G1F 5′-GGTAGCCTGGTGAAACATGG-3′ and SrUGT74G1R 5′CTGGGAGCTTTCCCTCTTCT-3′; and the primers corresponding to the glycosyltransferase *UGT76G1* gene were SrUGT76G1F 5′-GACGCGAACTGGAACTGTTG-3′ and SrUGT76G1R 5′-AGCCGTCGGAGGTTAAGACT-3′. qRT–PCR was performed using SYBR Green (QIAGEN, California, USA) and quantified on a Rotor-Gene Q (QIAGEN, California, USA) real-time PCR thermal cycler. qRT–PCR was programmed for 35 cycles, with denaturing at 95 °C for 15 s, annealing at 58 °C

for 30 s, and extension at 72 °C for 30 s. Three biological replicates with three technical replicates per treatment were evaluated. Primer specificity was verified by regular PCR and melting curve analysis. The primers for the *S. rebaudiana* glyceraldehyde-3-phosphate dehydrogenase (*GAPDH*) gene (SrGAPDHF 5′-TCAGGGTGGTGCCAAGAAGG-3′ and SrGAPDHR 5′-TTACCTTGGCAAGGGGAGCA- 3′) were used as internal controls for normalization, and the quantitative results were evaluated by the $2^{-\Delta CT}$ method described by *Livak & Schmittgen (2001)*.

## Statistical analysis

Raw data from each analysis were used to obtain the central tendency measures (means and standard deviations). Data on each parameter for the noninoculated and inoculated plants were analyzed by using one-way analysis of variance (ANOVA), and significant differences were analyzed using Tukey's test, with a *P* value < 0.05. All the data were checked for normality using Shapiro–Wilk's test before statistical analysis. All statistical analyses were performed using the statistical software Minitab® for Windows, version 19.1 (Minitab, LLC, State College, PA, USA), and figures were made using GraphPad Prism for Windows, version 6.0 (GraphPad Corp, San Diego, CA, USA).

## RESULTS

### Effect of endophytic bacterial inoculation on *S. rebaudiana* growth, steviol glycosides (SG), phenolic compounds, and flavonoid accumulation in the leaves

The inoculation with endophytic bacteria did not promote the growth of *S. rebaudiana* plants, since plant and root length, number leaves and root dry weight were not different from those of noninoculated plants (Table 2). However, the results of the effect of inoculation with endophytic bacteria on the accumulation of specialized metabolites, showed that there are stimulating endophytes (*Enterobacter hormaechei* H2A3, *E. hormaechei* H5A2) and nonstimulating endophytes. For the stimulating endophytes, it was observed that in plants inoculated with *Enterobacter hormaechei* H2A3, there was a significant increase in the concentrations of total SG (Fig. 1A), rebaudioside A (Fig. 1B), and stevioside (Fig. 1C), with values 2.2, 2.2 and 2.1- fold greater, respectively, than those in noninoculated plants. The same trend was found with the inoculation with *E. hormaechei* H5A2, where the concentrations of total SG, rebaudioside A, and stevioside increased significantly by 1.5, 1.5, and 1.4-fold, respectively, in comparison with those in the noninoculated plants. Inoculation with *E. bacterium* H7A1 did not significantly increase the concentration of specialized metabolites in comparison to noninoculated plants, but the concentration of metabolites was similar to that found in plants inoculated with *E. hormaechei* H5A2. Plants inoculated with other bacteria (nonstimulating endophytes) did not present significant changes in the concentration of specialized metabolites (Fig. 1).

In *S. rebaudiana* plants inoculated with *E. hormaechei* H5A2, there was a significant increase of 1.4-fold in the flavonoid concentration in comparison to that in noninoculated plants (Fig. 2A), while the concentration of phenolic compounds was similar to that in noninoculated plants (Fig. 2B). Inoculation with other bacteria did not increase
**Table 2 Growth parameters (length, number of leaves, leaf dry weight, root length and root dry weight) of *Stevia rebaudiana* plants inoculated with endophytic bacteria.** Data represent the mean ± standard deviation of ten individual plants harvested to 30 days postinoculation. The same letter indicates no statistically significant differences between growth parameters (*P* value of less than 0.05).

| Treatment | Plant length (cm) | Number of leaves | Leaves dry weight | Root length (cm) | Root dry weight |
|---|---|---|---|---|---|
| Control | 23.6 ± 2.5 a | 28.5 ± 3.3 a | 0.117 ± 0.03 a | 19.1 ± 2.2 a | 0.128 ± 0.05 a |
| *E. hormaechei* H2A3 | 19.7 ± 1.3 a | 27.2 ± 2.5 a | 0.082 ± 0.02 a | 18.9 ± 2.0 a | 0.116 ± 0.04 a |
| *E. hormaechei* H5A2 | 23.4 ± 1.5 a | 29.9 ± 3.7 a | 0.088 ± 0.01 a | 20.8 ± 2.4 a | 0.146 ± 0.04 a |
| *E. bacterium* H7A1 | 25.2 ± 7.3 a | 29.9 ± 3.7 a | 0.105 ± 0.05 a | 21.6 ± 1.2 a | 0.129 ± 0.07 a |
| *E. xiangfangensis* T1A2 | 21.9 ± 2.6 a | 28.6 ± 2.9 a | 0.103 ± 0.05 a | 23.9 ± 5.7 a | 0.107 ± 0.04 a |
| *E. xiangfangensis* T3A3 | 24.9 ± 3.1 a | 30.2 ± 3.5 a | 0.095 ± 0.02 a | 21.2 ± 2.9 a | 0.107 ± 0.05 a |
| *E. xiangfangensis* T5P1 | 25.8 ± 3.1 a | 31.0 ± 2.0 a | 0.108 ± 0.04 a | 21.1 ± 1.7 a | 0.125 ± 0.04 a |
| *E. xiangfangensis* R2A2 | 20.7 ± 2.1 a | 28.2 ± 1.9 a | 0.087 ± 0.03 a | 22.2 ± 4.7 a | 0.092 ± 0.03 a |
| *E. cloacae* R3A1 | 23.6 ± 1.7 a | 29.4 ± 0.7 a | 0.116 ± 0.03 a | 21.2 ± 2.7 a | 0.114 ± 0.04 a |
| *E. hormaechei* R5P1 | 24.5 ± 2.6 a | 28.6 ± 2.8 a | 0.099 ± 0.03 a | 23.0 ± 3.4 a | 0.104 ± 0.03 a |
| *E. hormaechei* R6A1 | 23.3 ± 3.7 a | 26.6 ± 5.0 a | 0.093 ± 0.01 a | 26.0 ± 6.3 a | 0.101 ± 0.02 a |
| *Bacillus safensis* R6P1 | 22.3 ± 1.3 a | 26.7 ± 1.8 a | 0.121 ± 0.02 a | 20.0 ± 2.3 a | 0.115 ± 0.02 a |
| *E. xiangfangensis* R7A2 | 23.3 ± 3.3 a | 29.7 ± 2.5 a | 0.087 ± 0.02 a | 21.6 ± 2.3 a | 0.102 ± 0.03 a |

the concentration of flavonoids and phenolic compounds, while inoculation with *E. xianfangensis* R7A2 significantly decreased the concentration of phenolic compounds.

Based on the screening results with endophytic bacteria, the selected bacteria used to continue this work were *E. hormaechei* H2A3 and *E. hormaechei* H5A2. Additionally, *E. xianfangensis* R7A2 was used as an additional treatment because it did not show induction of specialized metabolites or growth promotion. These bacteria were used to analyze the effect on trichome density in leaves as well as the expression of genes of the SG biosynthesis pathway.

## Trichome density in *S. rebaudiana* leaves by scanning electron and confocal microscopy

Photomicrographs (SEM) of the leaves from noninoculated plants and plants inoculated with the selected bacteria showed three types of trichomes: glandular, large, and short (Fig. 3, see labels G, L and S). In plants inoculated with *E. hormaechei* H2A3, the photomicrographs showed the presence of a greater number of trichomes that were short, large and glandular in comparison with those in noninoculated plants (Fig. 3B, see labels S, L and G), while in leaves of plants inoculated with *E. hormaechei* H5A2 and *E. xiangfangensis* R7A2, the photomicrographs did not show a visual effect on the trichome number in relation to that in noninoculated plants (Figs. 3C–3D).

In the leaves of noninoculated and inoculated plants, short trichomes were the most abundant (2,000 to 6,000 trichomes/cm$^2$), followed by glandular (1,000 to 3,000 trichomes/cm$^2$) and large trichomes (200 to 800 trichomes/cm$^2$) (Fig. 4). Trichome density showed that *E. hormaechei* H2A3 induced a significant increase in glandular, large, and short trichomes of 1.7, 4.3, and 1.5-fold, respectively, in comparison to those in noninoculated plants (Figs. 4A–4C). However, *E. hormaechei* H5A2 did not induce any

<br>

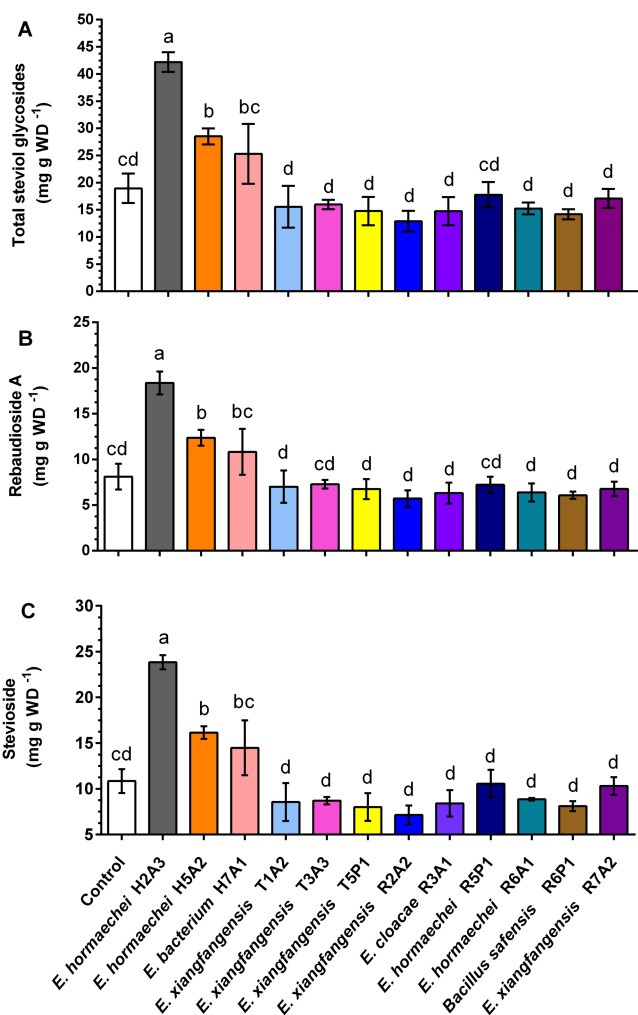

**Figure 1** **Content of total steviol glycosides (A), stevioside (B) and rebaudioside A (C) in *Stevia rebaudiana* leaves of noninoculated (control) and inoculated plants with endophytic bacteria.** Bars represent the mean ± standard deviation of three individual plants collected to 30 days postinoculation. Different letters indicate a significant difference according to Tukey's test (*P* value of less than 0.05).

effect on trichome density (Figs. 4A–4C) and *E. xiangfangensis* R7A2 had significantly lower number of short, and did not induce effect on glandular and large trichomes compared to control (Figs. 4A–4C).

The location of SG, phenolic compounds, and flavonoids in the trichomes of *S. rebaudiana* leaves by autofluorescence using confocal microscopy is shown in Fig. 5. In the red channel, the autofluorescence of chlorophylls is shown in epidermal and mesophyll leaf cells (Figs. 5A–5D), while in the blue channel, the autofluorescence of SG, phenolic compounds, and flavonoids is shown in the trichomes (Figs. 5E–5H). Inoculation with *E. hormaechei* H2A3 and *E. hormaechei* H5A2 increased the intensity of the autofluorescence signal in the blue channel, particularly in glandular and short trichomes (Figs. 5F and 5G, see labels G and S), while in the noninoculated plants (Fig. 5E) and those inoculated with

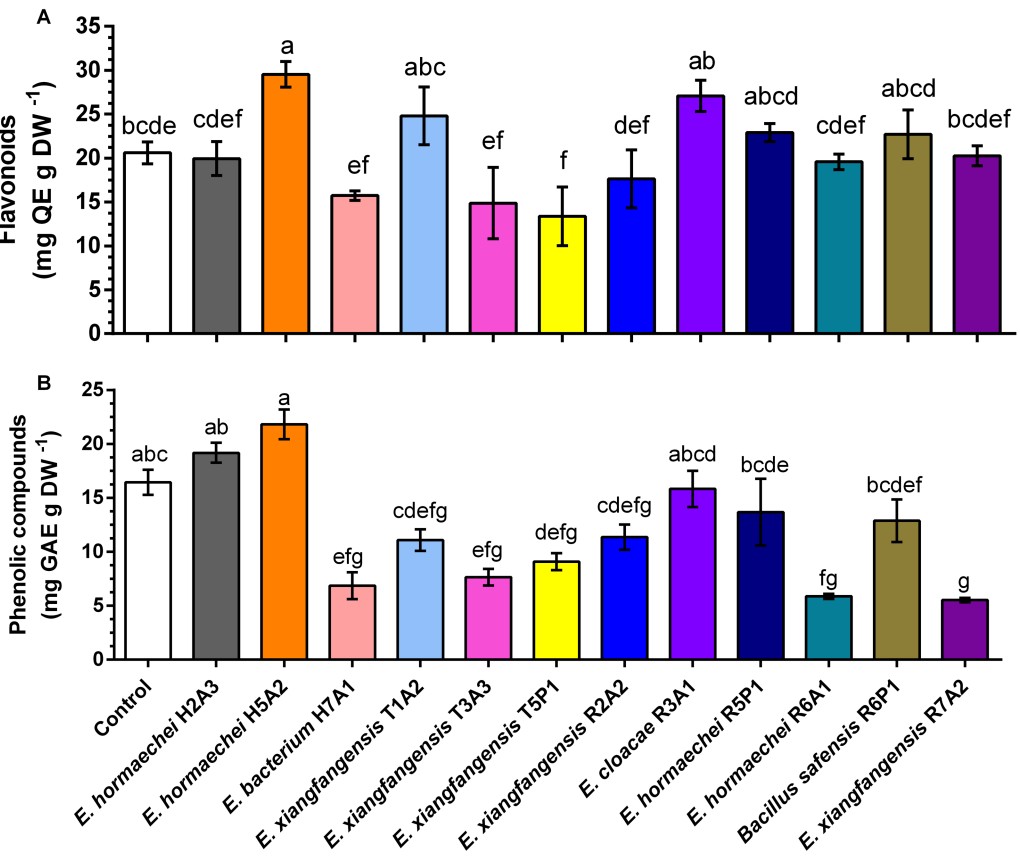

**Figure 2** **Content of flavonoids (A) and phenolic compounds (B) in *Stevia rebaudiana* plants inoculated with endophytic bacteria.** Bars represent the mean ± standard deviation of three individual plants collected to 30 days postinoculation. Different letters indicate a significant difference according to Tukey's test (*P* value of less than 0.05).

*E. xiangfangensis* R7A2 (Fig. 5H, see labels L, G, and S), the autofluorescence signal was lower.

Notably, strong blue fluorescence was exhibited in the secretory cavity of the glandular trichomes and on the short trichomes of inoculated plants with *E. hormaechei* H2A3 and *E. hormaechei* H5A2 (Figs. 5N and 5O, see labels S and G) in comparison to the noninoculated plants (Fig. 5M) and those inoculated with *E. xiangfangensis* R7A2 (Fig. 5P).

## Effect of endophytic bacterial inoculation on differential SG biosynthetic gene expression in *S. rebaudiana* plants

The results of the differential expression of SG biosynthesis genes in *S. rebaudiana* plants inoculated with endophytic bacteria are presented in Fig. 6. The transcription level of the *KO* gene increased significantly with *E. hormaechei* H5A2 bacteria (21.3-fold) and *E. xiangfangensis* R7A (42.3-fold) compared to that inoculated with *E. hormaechei* H2A3 and that in noninoculated plants (Fig. 6A). The *KAH* transcript level increased significantly by 52.3-fold with *E. hormaechei* H5A2 (Fig. 6B). The transcript levels of the *UGT74G1* gene were significantly increased with inoculation with *E. hormaechei* H2A3 (11.3-fold) and

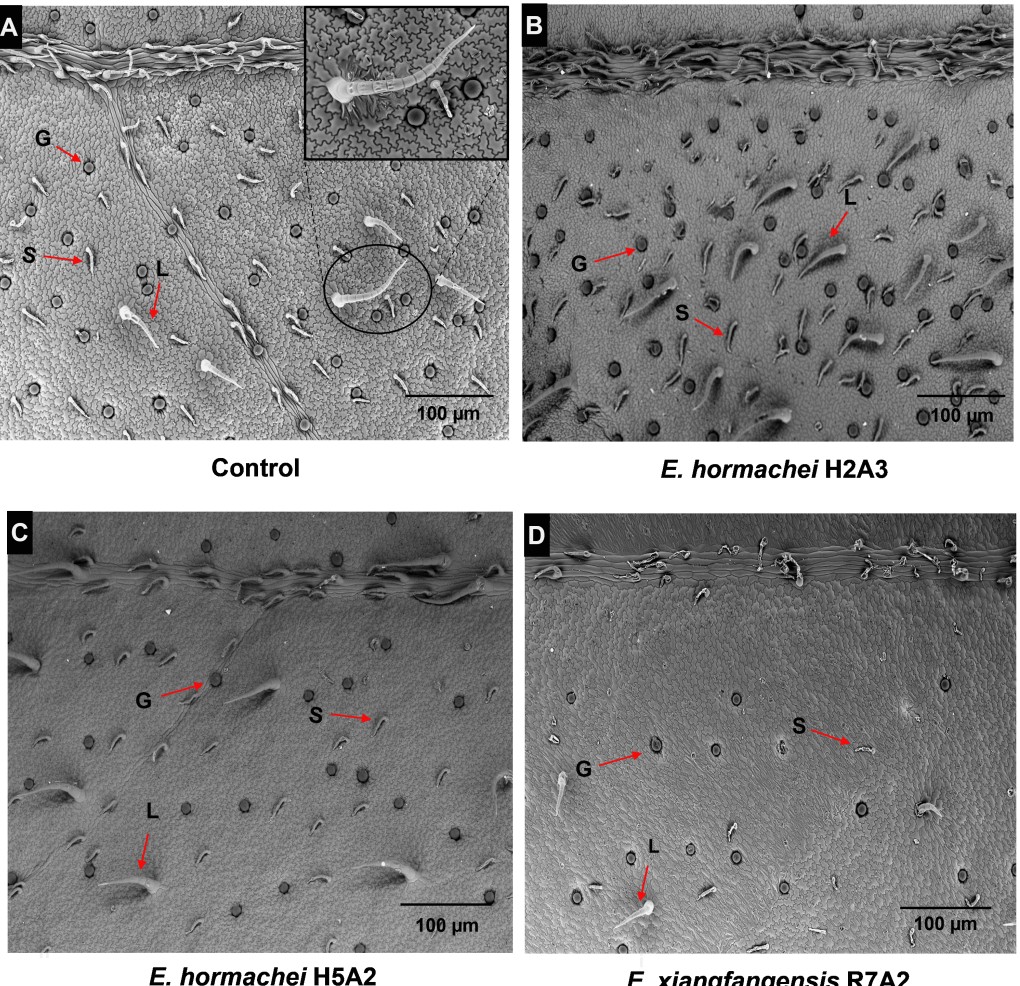

**Figure 3** **Photomicrographs obtained with environmental scanning electron microscopy of trichomes in *Stevia rebaudiana* leaves.** Trichomes in leaves of (A) noninoculated and inoculated plants with (B) *E. hormaechei* H2A3, (C) *E. hormaechei H5A2* and (D) *E. xiangfangensis* R7A2. 4x magnified Panel A corresponds to the image of trichome morphology of the uninoculated leaves. In all panels, the letter "S" is used to indicate short trichomes, "L" to indicate large trichomes and "G" to indicate glandular trichomes.

*E. hormaechei* H5A2 (17.2-fold), while *E. xiangfangensis* R7A increased (6.0-fold) but was not significant with respect to that in noninoculated plants (Fig. 6C). Finally, the transcript levels of the *UGT76G1* gene were significantly increased by 3.2-fold with the addition of *E. hormaechei* H2A3 (Fig. 6D).

# DISCUSSION

Interactions between plant and endophytic microorganisms have been proposed as a strategy to improve plant growth and stimulate secondary metabolism (*Afzal et al., 2019*; *Hardoim et al., 2015*; *Hardoim, Van Overbeek & Van Elsas, 2008*). However, in this work, reinoculation with endophytic bacteria isolated from different tissues of *S. rebaudiana* did

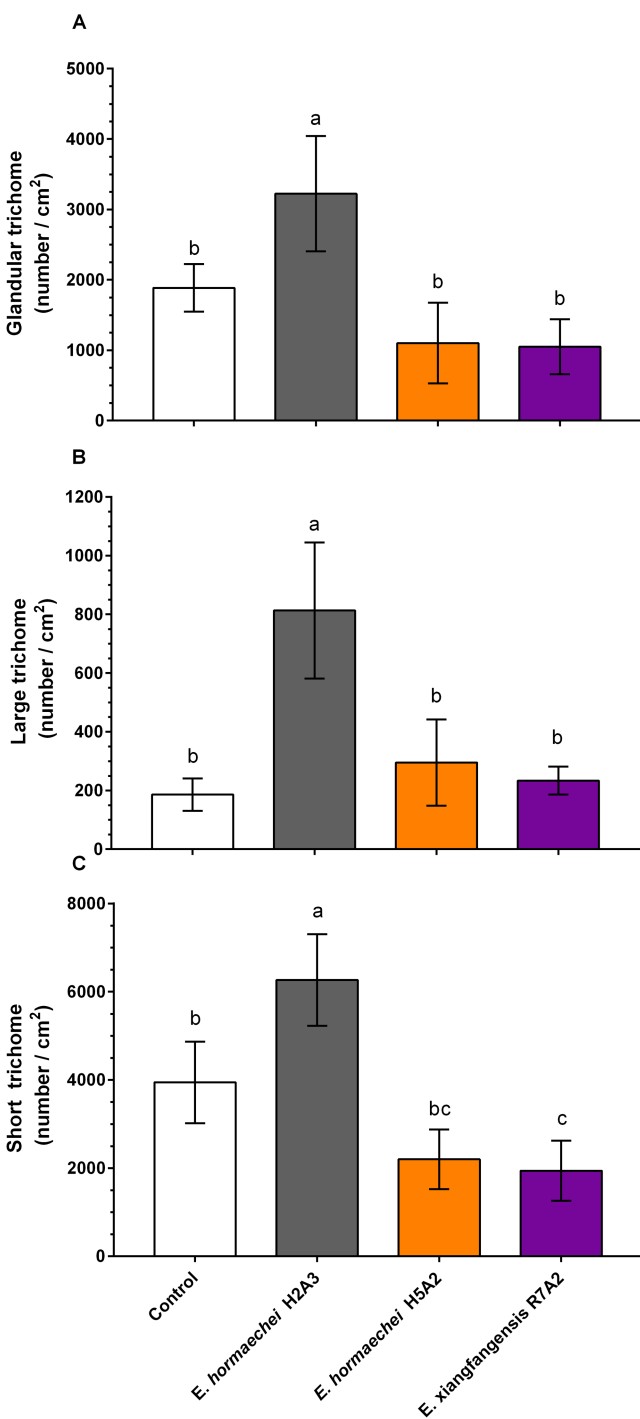

**Figure 4** Effect of endophytic inoculation on the trichome density of *Stevia rebaudiana* leaves. (A) **Glandular, (B) large, and (C) short trichome density in leaves.** Bars represent the mean ± standard deviation of four individual plants collected 30 days postinoculation. The different letters indicate a significant difference according to Tukey's test ($P$ value of less than 0.05).

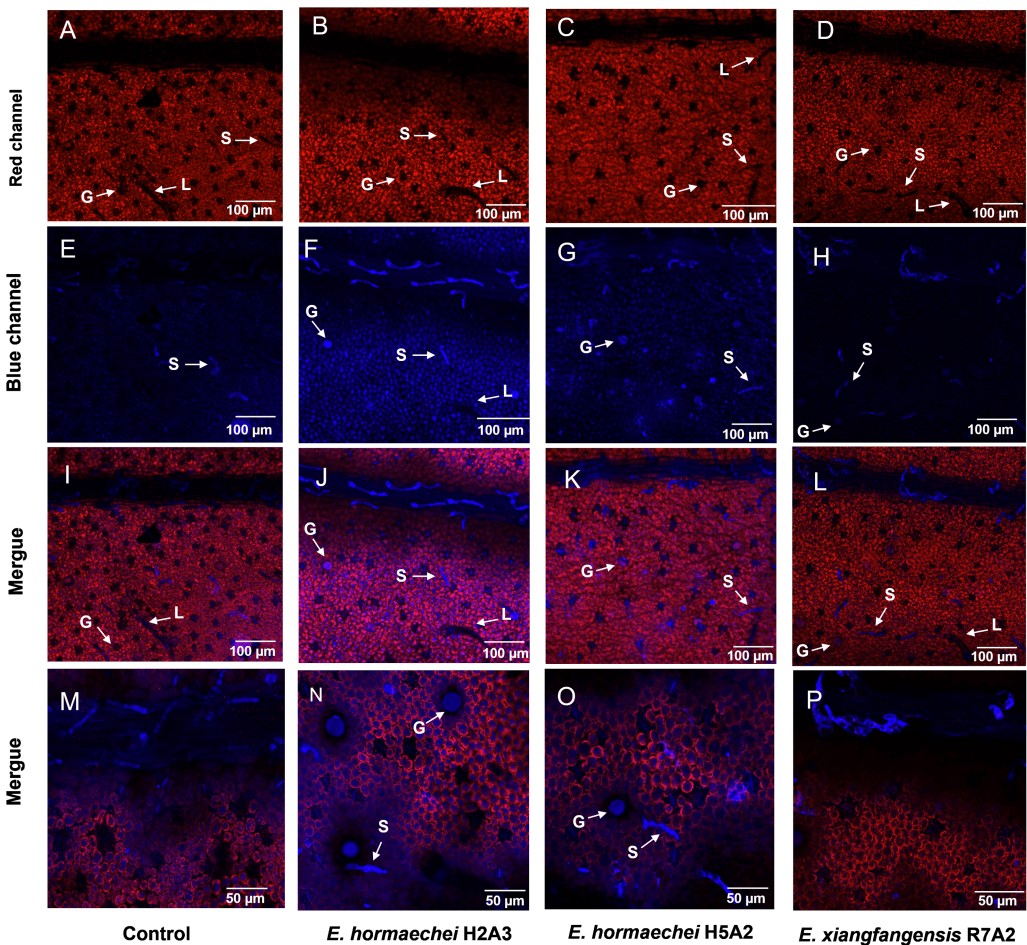

**Figure 5  Localization of the specialized metabolites steviol glycosides, phenolic compounds, and flavonoids in *Stevia rebaudiana* trichomes.** Chlorophyll autofluorescence (red channel) of (A) noninoculated and inoculated plants with (B) *E. hormaechei* H2A3, (C) *E. hormaechei* H5A2, and (D) *E. xiangfangensis* R7A2. Specialized metabolite autofluorescence (blue channel) of (E) noninoculated and inoculated plants with (F) *E. hormaechei* H2A3, (G) *E. hormaechei* H5A2, and (H) *E. xiangfangensis* R7A2. Colocalization of chlorophyll and specialized metabolite autofluorescence in (I) noninoculated and inoculated plants with (J) *E. hormaechei* H2A3, (K) *E. hormaechei* H5A2, and (L) *E. xiangfangensis* R7A2. Panels M-P correspond to a magnification of trichomes in noninoculated and inoculated plants with *E. hormaechei* H2A3, *E. hormaechei* H5A2, and *E. xiangfangensis* R7A2, respectively. The chlorophyll and specialized metabolite fluorescence emission spectra were obtained at 630–685 nm (red channel) and 435–485 nm (blue channel), respectively. In all panels, the letter "S" is used to indicate the shot trichomes, "L" to indicate the large trichomes and "G" to indicate the glandular trichomes.

not significantly promote plant growth. These results suggest that growth promotion is not associated with endophytic bacterial reinoculation and that the bacteria did not negatively affect plant growth. It is possible that the plant could divide the nutrients for primary metabolism or provide the nutrients required for bacterial growth. This behavior has also been observed in different plant–microorganism interactions, such as *Ocimum basilicum* L. inoculated with *Glomus mosseae* (*Copetta, Lingua & Berta, 2006*) and *Ocimum gratissimum* L. inoculated with *Glomus intrarradices* (*Hazzoumi, Moustakime & Joutei, 2017*).

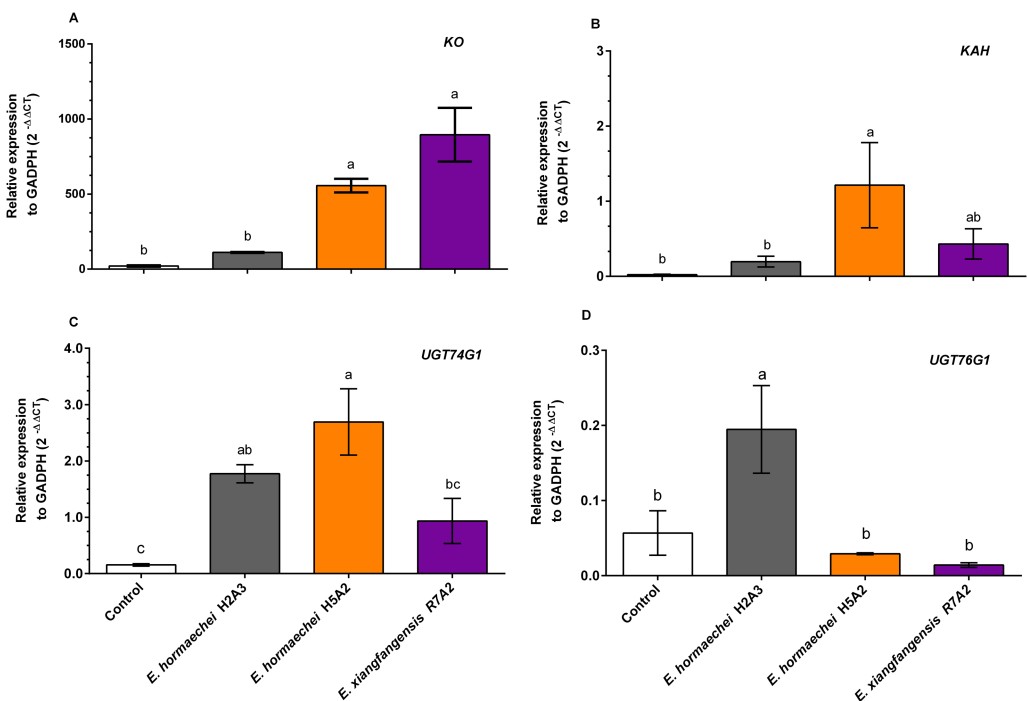

**Figure 6** Relative expression analysis of key genes in the steviol glycoside biosynthesis pathway of *Stevia rebaudiana* leaves noninoculated and inoculated with endophytic bacteria. (A) *KO*, (B) *KAH*, (C) *UGT74G1*, (D) *UGT76G1*. The relative expression of *GAPDH* was calculated. The $2^{-\Delta CT}$ method was used. Bars represent the mean $\pm$ standard deviation of three biological and three technical replicates collected 30 days postinoculation. Different letters indicate significant differences according to Tukey's test ($P$ value of less than 0.05).

The stimulating endophytes *E. hormaechei* H2A3 and *E. hormaechei* H5A2 increased the concentrations of stevioside and rebaudioside A in the leaves of *S. rebaudiana*. *E. hormaechei* H5A2 increased the concentration of flavonoids, which indicates that these bacteria play an important role in the biosynthesis of the specialized metabolites, SG and flavonoids in *S. rebaudiana*. Similarly, other studies have shown that bacteria may have a differential effect on the biosynthesis of secondary metabolites in crops such as *Oryza sativa* L. (*Andreozzi et al., 2019*; *Balachandar et al., 2006*), *Beta vulgaris* L. (*Shi, Lou & Li, 2010*), *Artemisia annua* L. (*Li et al., 2012*; *Tripathi et al., 2020*), *Catharanthus roseus* (Nirmal) (*Tiwari et al., 2013*), *Salvia miltiorrhiza* Bunge (*Yan et al., 2014*), *Fragaria ananassa* (Duch) Macarena (*Guerrero-Molina et al., 2014*), *Glycine max* (L.) Merr (*Asaf et al., 2017*), *Glycyrrhiza uralensis* F (*Li et al., 2018*), *L. radiata* (*Liu et al., 2020*), and *Camellia oleifera* Abel (*Xu et al., 2020*). The effect of inoculation with bacteria and fungi on the growth and synthesis of metabolites in *S. rebaudiana* plants has been reported. *Vafadar, Amooaghaie & Otroshy (2014)* reported that bacteria isolated from the rhizosphere (*Bacillus polymixa*, *Pseudomonas putida* and *Azotobacter chroococcum*) inoculated in *S. rebaudiana* plants significantly increased root and shoot biomass and the concentrations of stevioside, chlorophyll, and macronutrients (nitrogen, phosphorus and potassium) in plants. *Kilam et al. (2015)* reported that the bacterium *A. chroococcum* improved the growth, antioxidant

activity and SG content of *S. rebaudiana* plantlets grown *in vitro*. Several fungi, including *G. intraradices, Piriformospora indica, Rhizoglomus irregulare*, and *Rizophagus intraradices*, have been reported as other inoculant microorganisms of *S. rebaudiana*, and several results have demonstrated that these fungi can enhance plant growth and stevioside accumulation (*Kilam et al., 2015; Mandal et al., 2013; Mandal et al., 2015; Vafadar, Amooaghaie & Otroshy, 2014; Sarmiento-López et al., 2020; Tavarini et al., 2018*). A synergistic relationship between bacteria and fungi has been proposed to improve the plant growth of *S. rebaudiana* and the accumulation of SG (*Kilam et al., 2015; Vafadar, Amooaghaie & Otroshy, 2014*). *Nowogórska & Patykowski (2015)* findings support the idea that sequential inoculation with bacteria, fungi, or a combination of both does not always yield synergistic effects. However, this issue should be investigated in the future. To our knowledge, this is the first report of inoculation of *S. rebaudiana* with endophytic bacteria from the *Enterobacter* genus as a strategy to improve the biosynthesis of their specialized metabolites. The results of our study indicate that the synthesis of specialized metabolites is achieved with the inoculation of endophytic bacteria without fungal coinoculation.

Trichomes are plant structures that accumulate secondary metabolites, and their presence in plant leaves is associated with defense mechanisms of the plant against pathogens, insects, and adverse environmental conditions (*Champagne & Boutry, 2016; Tian et al., 2017; Werker, 2000*). The trichomes observed in the leaves of *S. rebaudiana* were short, large, and glandular. This trichome morphology was typical of those previously reported (*Bondarev et al., 2003; Bondarev et al., 2010; Cornara et al., 2001; Monteiro et al., 2001*). Our results showed that inoculation with endophytic bacteria caused a significant increase in trichome density in *S. rebaudiana* leaves. This anatomical response in the increased trichome density has been observed in other plants, such as *A. annua* inoculated with *R. intraradices* (*Mandal et al., 2015*) and *A. annua* inoculated with *Glomus macrocarpum* and *Glomus fasciculatum* (*Kapoor, Chaudhary & Bhatnagar, 2007*). Inoculation with the endophyte *E. hormaechei* H2A3 generated a higher density of trichomes in *S. rebaudiana* leaves as well as a higher concentration of SG and flavonoids in comparison to the control. However, the results of a Pearson analysis between the concentration of specialized metabolites and trichome density did not show a correlation between the variables ($R^2 < 0.53$). These results are in contrast with *Bondarev et al. (2010)*; the authors suggest a positive relationship between the number of glandular trichomes and the accumulation of SG; however, they did not present a quantitative analysis of the correlation between SG accumulation and trichome density.

In other plants that accumulate secondary metabolites in trichomes, a relationship between the number of trichomes in the leaves and the accumulation of secondary metabolites induced by inoculation with different fungi was reported. *Kapoor, Chaudhary & Bhatnagar (2007)* and *Mandal, Upadhyay & Wajid (2015)* described that the inoculation of beneficial fungi (*Glomus macrocarpum, Glomus fasciculatum* and *Rhizophagus intraradices*) in *A. annua* plants enhanced the accumulation of artemisinin in trichomes and reinforced the idea that beneficial interactions, including endophytic bacteria, induce several biochemical and physiological responses for the benefit of crops.

The use of confocal microscopy tools used in this work allowed the localization of the specialized metabolites in the trichomes of *S. rebaudiana* leaves by detecting their autofluorescence (*Agati et al., 2002*; *Talamond, Verdeil & Conéjéro, 2015*; *Vidot et al., 2019*). In this work, autofluorescence in the blue spectrum, which is indicative of the accumulation of these metabolites, was found in trichomes of inoculated *S. rebaudiana* by *E. hormaechei* H2A3 and *E. hormaechei* H5A2. In plants, the accumulation of different specialized metabolites has been observed in trichomes (*Agati et al., 2002*; *Conéjéro et al., 2014*; *Hutzler et al., 1998*; *Talamond, Verdeil & Conéjéro, 2015*). Recently, *Sarmiento-López et al. (2021)* reported that *S. rebaudiana* plants colonized with arbuscular mycorrhiza fungi *R. irregularis* showed fluorescence in the trichomes and that this was related to the increase in phenolic compounds and flavonoid accumulation. Taken together, these results suggest that a similar mechanism for metabolite induction and accumulation occurs in both endophytic bacterial and fungal interactions with plants. It is well known that specialized metabolites such as terpenes and phenolic compounds have an important function in the priming response by activating systemic resistance, enabling plants to respond more effectively to attacks from pathogens and herbivores (*Cervantes-Gámez et al., 2016*; *Pozo & Azc'on-Aguilar, 2007*; *Santos, Alves da Silva & Barbosa da Silva, 2017*). In this work, we observed a significant increase in trichome development, as well as in the accumulation of SG and phenolic compounds, which can be related to the induction of systemic resistance by the inoculation of endophytic bacteria in *S. rebaudiana* plants in a manner similar to that observed in other plant species under different plant–microorganism interactions (*Kapoor, Chaudhary & Bhatnagar, 2007*; *Mandal et al., 2015*). However, in *S. rebaudiana*, further experimental studies are needed to prove this hypothesis.

Kaurene oxidase and kaurenoic acid hydroxylase are important enzymes in SG biosynthesis and represent the principal branch point in the catabolism of the central backbone (steviol) of SG. In fact, steviol is glycosylated by the conjugation of glucose by UDP-glycosyltransferases (*UGTs*), where *UGT74G1* is responsible for synthesizing stevioside, while *UGT76G1* is required to produce rebaudioside A (*Brandle & Telmer, 2007*). Our results of gene expression analysis of the SG biosynthesis pathway in *S. rebaudiana* leaves showed that the *KO* gene was upregulated with *E. hormaechei* H5A2 and *E. xiangfangensis* R7A2; the *KAH* gene was upregulated with *E. hormaechei* H5A2. Likewise, the *UGT74G1* gene was upregulated with the inoculation of *E. hormachei* H2A3 and *E. hormachi* H2A3 and *E. hormaechei* H5A2, which was consistent with the high stevioside concentration, whereas the *UGT76G1* gene was upregulated with *E. hormachei* H2A3 inoculation, which may be directly related to the rebaudioside A concentration determined in *S. rebaudiana* leaves. Inoculation with *E. hormachei* H5A2 also stimulated rebaudioside A accumulation, but it was not reflected in the expression of genes involved in their metabolite synthesis. Although the transcript levels in plants inoculated with *E. hormachei* H5A2 were low, it is possible that the enzymatic activity of (UDP)-glycosyltransferases synthetized by the *UGT76G1* gene could be similar to that in plants inoculated with *E. hormachei* H2A3. However, further complementary studies of enzymatic activity are necessary to confirm this hypothesis.

Previously, other rhizospheric microorganisms inoculated in *S. rebaudiana* plants showed improved SG accumulation, and the effect was associated with the high expression of their biosynthesis genes, *KO, KS, KHA, UGT74G1* and *UGT76G1* (*Kilam et al., 2015*; *Mandal et al., 2013*; *Tavarini et al., 2018*; *Vafadar, Amooaghaie & Otroshy, 2014*). In other plants, inoculation with endophytic bacteria also increased the content of secondary metabolites and the expression of genes in their biosynthetic pathway. For example, *Pseudonocardia* species induce the production of artemisinin in *A. annua* (*Li et al., 2012*), and *Acinetobacter* sp. induces abscisic acid (ABA) and salicylic acid (SA) production in *Atractylodes lancea* (Thunb.) DC. (AL) (*Wang et al., 2014*). The findings of the present work show that the use of the endophytic bacteria *E. hormachei* H2A3 and *E. hormachei* H5A2 can be considered a biotechnological strategy to increase the concentration of specialized metabolites in *S. rebaudiana*.

## CONCLUSIONS

Endophytic bacteria inoculated in *S. rebaudiana* plants did not promote plant growth, but the stimulating endophytes *E. hormaechei* H2A3 and *E. hormaechei* H5A2 increased the SG content and stimulated the density of trichomes in the leaves, as well as the accumulation of specialized metabolites in trichomes. The increase in the transcript levels of the *KO, KAH, UGT74G1*, and *UGT76G1* genes was correlated with SG concentration by inoculation with *E. hormaechei* H2A3 and *E. hormaechei* H5A2. These results suggest the potential use of *E. hormaechei* H2A3 and *E. hormaechei* H5A2 to increase the content of SG and flavonoids in *S. rebaudiana* plants.

## ACKNOWLEDGEMENTS

We owe special thanks to Dr. Luis Cardenas Torres from Instituto de BiotecnologÃa (Universidad Nacional Autónoma de México) for access to the laboratory to perform molecular biology tests of *S. rebaudiana* tissue. The authors are grateful to Msc. Daniel Tapía Maruri for providing technical assistance in producing the environmental scanning electron microscope and confocal laser.

### Funding

The authors received funding from the Consejo Nacional de Ciencia y Tecnología (Curriculum Vitae Unico: 702975) and Secretaria de Investigación y Posgrado-Instituto Politécnico Nacional (Beca de Estimulo Institucional de Formación de Investigadores). This work was conducted with the support of Secretaria de Investigación y Posgrado-Instituto Politécnico Nacional (projects: 20210733 and 202200672). The funders had no role in study design, data collection and analysis, decision to publish, or preparation of the manuscript.

### Grant Disclosures

The following grant information was disclosed by the authors:

Consejo Nacional de Ciencia y Tecnología: 702975.

Secretaria de Investigación y Posgrado-Instituto Politécnico Nacional: 20210733, 202200672.

## Competing Interests

The authors declare there are no competing interests.

## Author Contributions

- Dumas G. Oviedo-Pereira conceived and designed the experiments, performed the experiments, analyzed the data, prepared figures and/or tables, and approved the final draft.
- Melina López-Meyer conceived and designed the experiments, analyzed the data, prepared figures and/or tables, authored or reviewed drafts of the article, and approved the final draft.
- Silvia Evangelista-Lozano performed the experiments, analyzed the data, authored or reviewed drafts of the article, and approved the final draft.
- Luis G. Sarmiento-López conceived and designed the experiments, performed the experiments, prepared figures and/or tables, and approved the final draft.
- Gabriela Sepúlveda-Jiménez conceived and designed the experiments, analyzed the data, authored or reviewed drafts of the article, and approved the final draft.
- Mario Rodríguez-Monroy conceived and designed the experiments, analyzed the data, prepared figures and/or tables, authored or reviewed drafts of the article, and approved the final draft.

## Data Availability

The raw data is available in the Supplemental Files.

## Supplemental Information

Supplemental information for this article can be found online at http://dx.doi.org/10.7717/peerj.13675#supplemental-information.

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
