# Peer review of "Enhanced specialized metabolite, trichome density, and biosynthetic gene expression in Stevia rebaudiana (Bertoni) Bertoni plants inoculated with endophytic bacteria Enterobacter hormaechei"

_PeerJ, doi:10.7717/peerj.13675_

## Round 0.1 · original submission · Major Revisions

I think all three reviewers gave good suggestions and I hope you will make serious revisions to the article, both in terms of content and writing. Reviewer 2 also gave suggestions in the "annotated manuscript", some of which also require "point-to-point" responses.

Reviewer 1 ·

Basic reporting

The present manuscript reports the effects of inoculation with endophytic bacterial isolates on the growth and specialized metabolism of Stevia rebaudiana (Bertoni) Bertoni. I think that the Authors studied an important topic using adequate methods. The structure of the presented manuscript is typical for research article. Background is well described, methods are clear (with some exceptions) and data is presented concisely. My major comments pertain to the lines 140-141, 145-146 and 169-170 (methods). These issues need detailed explanation. Aim of the study is clear but 2-3 hypotheses must be stated. Some fragments in Discussion section must be reworked (as they can be more in-depth). Additionally, the Authors can consider to conduct more statistical analyses (this is not mandatory in my opinion and I feel that this is for the decision of the Editor). The language needs slight polishing but article is readable. The raw data is shared (but need clarification as mentioned above), and figures are of good quality. I believe that this article can be published after revision.

Experimental design

The study falls within the scope of PeerJ. The aim is clear, however, the hypotheses must be stated (see detailed list of comments). Methods are clear with some exceptions (see list of comments).

Validity of the findings

In my opinion the findings are validate, however, they can be improved as there were some purely descriptive fragments in the Discussion. In such case, some ideas for the Authors were proposed. The authors can try to do some more statistical analyses in order to see wider image of this study (e.g., correlation of the traits, differences between inoculants), but this is optional, not mandatory.

Additional comments

Request for global changes and improvements:
The term “secondary metabolites” was used for a very long time, but the term “specialized metabolites” has been used more often recently, as it is not known if these compounds are of secondary importance. I believe that this change is not mandatory but very desirable; you can consider such change.

Please always use full species name when introducing them for first time, e.g., Lycoris radiata (L'Hér.) Herb. but not Lycoris radiata. Please check it and change where applicable for other species. You can use the Plant List database or World Flora Online database.

Species naming needs more uniformity. When you introduce name of species, e.g., Stevia rebaudiana (Bertoni) Bertoni, give the full name and, then, use consequently stevia (common name) or S. rebaudiana (short species name) but do not mix these conventions throughout the manuscript (as in lines 61 and 62). Both forms are fully acceptable, however, please select one of them. Additional comment: APG IV classification states that the correct name of the studied species is Stevia rebaudiana (Bertoni) Bertoni. Thus, please use this name or cite reference for Stevia rebaudiana Bertoni if there is a rationale.

I believe that names of proteins (and nonprotein metabolites as well), including enzymes should be presented without italics, while names of their genes should be italicized. Please check it carefully throughout the manuscript, from line 65. For example, it is done very well in line 90, but in L66-73 needs some attention.

As this work describes biochemical analyses, I believe that SI units should be used, e.g., cm3 but not mL and dm3 but not L, etc. Additionally, please replace mg/g in the dataset for mg g-1 as you did in the figures. Please check and change where applicable.

Were the leaves that you used for quantification of metabolites fully developed? Were they from upper or lower part of the plant? Please provide this information when you describe procedure for extraction.
Please provide EC numbers of enzymes that you describe and study.

Request on statistical analysis and data collection: I wonder that for some measurements you consider only control, H2A3-, H5A2- and R7A7-inoculated plants because control is always needed, H2A3-, H5A2-inoculated plants showed the best induction of steviol glycosides biosynthesis but why R7A7-inoculated plants? Was there any rationale? If was this the group with the lowest performance, that should be clearly showed in Figure 1 and Figure 2. Additionally, more information can be extracted from this dataset. Dunnett’s post-hoc test was used to compare treated variants with control variant (it is the correct approach), but comparison among all the studied variants is also very valuable. Were there any differences between H2A3-, H5A2-inoculated plants? How about other treatments? This can be analyzed using other post hoc tests which I strongly recommend. Then, the selection of the R7A7 variant is fully justified. Was there any other rationale? This change is not mandatory but 1) there is chance to introduce statistical error due to approach you used, and 2) your findings can be misinterpreted by the audience. I understand that conducting statistical analysis from scratch is time-consuming, but in my opinion this is very desired for this dataset. Additionally, did you consider correlation analysis? Maybe there is correlation between number of trichomes and contents of the studied metabolites (at least in some cases)?

Figure 1: Stevioside and rebaudioside A have the same color-based annotations. The use of color artwork in the PeerJ is free of change; thus, you can use color to correct this issue.

Figure 3: Size should be greater (see detailed list of comments).

Detailed list of issues:
L41: You can consider to introduce the information that plant-bacteria associations can be phyllospheric, rhizospheric and endophytic.
L52: I think that capital letter should there used, as name of taxon is presented: Amaryllidaceae alkaloid (like in eponyms).
L55: Please remove italics from “Bertoni”.
L56: I believe that it can be stated that Stevia rebaudiana (…) is an economically important crop due to its ability to accumulate specialized metabolites called steviol glycosides (SG), including isosteviol, stevioside, rebaudiosides (A, B, C, D, E and F), steviolbioside and dulcoside A that are used as low-calorie sweeteners (…).
L59: Capital letter should not be used for stevia common name in the middle of the sentence (please check where applicable).
L77-79: These two sentences can be rewritten to improve their structure, e.g.: In S. rebaudiana, the presence of both short and long glandular trichomes has been reported (Sarmiento-López et al., 2021). The following fragment must be corrected: “they increase in leaves by arbuscular mycorrhizal colonization, as well as the accumulation of phenolic compounds”. Does it pertain to their number or size? Additionally, their size/number increase due to enhanced specialized metabolism, or content of specialized metabolites increase due to mycorrhizis?
L88: The meaning of AM should be introduced.
L95: The aim of the study is very well presented, but this is the best place to state your hypotheses. Do you expect improved biosynthesis of the studied metabolites as the result of inoculation? Can it be hypothesized a priori that the studied bacterial isolates will trigger different (or the same response)? Will there be any trade-offs between plant growth per se and biosynthesis or both growth and biosynthesis be improved? Please present here 2-3 short hypotheses that piqued your curiosity about this topic.
L100-105: You refer to the previous work where cultivation of plants was described in detail, but please add detailed information about growth conditions here as well. It will improve the readability. Additionally, were these cultures centrifuged+washed before incolutaion of plants?
L107: Please describe in detail procedure of isolation of endophytic bacteria, e.g., source of donor plants, media used for isolation, etc. Did you use all isolated strains or were there any additional criteria for their selection?
L115: Were the plants sown or planted?
L124: Height measurements: was initial height of the plants subtracted from their final height? Were they planted the same way to avoid subtraction?
L126: Please correct, I propose: (…) to determine their dry weight
L136: If possible, please provide also g value for centrifugation
L140-141: You used six plants per treatments and two replications but in the dataset I can see only three values per treatment. Were the samples pooled? It is often used method to reduce the costs or time, and in my opinion is acceptable, but it must be described in detail.
L145-146: It must be clarified. Did you mean that 0.1 g of sample was extracted in 100 cm3 of ethanol when you stated 0.1%? You can also state that the sample to fluid ratio was 1:1000.
L152: Please provide concentration range of the calculated standard curve and R^2 value (proof of linearity).
L158: Please provide concentration range of the calculated standard curve and R^2 value (proof of linearity).
L164: Were the leaves you used fully developed?
L169-170: You stated that counting of trichomes was performed according to the Sarmiento-López et al. (2021). However, this work does not contain details on differentiation between long and short trichomes. I tried to identify them on Figure 3 and this is easy and possible, however, I recommend to organize this figure in new manner. Instead of 1x5 figure (where all photographs are small) use 3+2 or 2+2+1 to make these image larger. The audience must see that short trichomes were short and long were long. Were they counted on whole area of 0.2826 cm2 (as in the Sarmiento-López et al., 2021)? It must be stated in detail.
L207-208: Needs language correction.
L212-288: Please always use term “significantly” when you state the differences, e.g., significantly lower, significantly higher, significantly different.
L212-288: If the difference were significant, always state their magnitude, e.g., 1.65 times more, 65% more, 165% of the control variant, etc.
L283-288: This needs attention, as in the Results section you should not reference to the previous studies.
L291: Plant–endophytic
L293-294: Did you mean screening/isolation of endophytic bacterial isolates and their reinoculation? Please rewrite this sentence.
L294-295: You can also state that inoculation did not halt nor reduce plant growth. It is common that plants must divide their resourced into growth of comping with stimuli. In my opinion, it is very important information that yield quality was improved whilts its quantity was not reduced. It can also be another sign that the studied isolates were endophytic but not random or facultative pathogenic.
L298: in this context, “triggered” is probably better word than “produced”.
L305: Fragaria ananassa but not Fragaria ananassia. Please also verify other plant names.
L313: Please change NPK to “nitrogen, phosphorus and potassium” or “macronutrients (nitrogen, phosphorus and potassium”.
L323-325: This can be stated better: “(…) the accumulation of SG (Kilam et al., 2015; Vafadar, Amooaghaie & Otroshy, 323 2014). However, the present work reports that stimulation of SG biosynthesis can be achieved with selected endophytic bacteria without fungal coinoculation. Although sequentional incoluation with bacteria and fungi does not always yield synergistic effects (Nowogórska and Patykowski, 2015), this issue should be investigated in the future because plants are exposed to numerous environmental stimuli acting simultaneously).” (Nowogórska A, Patykowski J. 2014. Selected reactive oxygen species and antioxidant enzymes in common bean after Pseudomonas syringae pv. phaseolicola and Botrytis cinerea infection. Acta Physiologiae Plantarum 37. DOI: 10.1007/s11738-014-1725-3.
L329-331: You can only state that morphology of trichomes was typical for the studied species.
L333-334: Statement about coincidence of traits should be supported by correlation analysis. Please rewrite this or add additional results pertaining to the analysis of correlation.
L345-361: I feel this fragment too descriptive. I think that you can state that improved contents of the studied metabolites was associated with greater autofluorescence of trichomes and, based on the available data, improved biosynthesis in trichomes is at least partially responsible for the observed effect. Then, it can be discussed in a practical context and through the prism of ecological implications. Is it fully beneficial for plants to contain significantly increased contents of specialized metabolites? How about tradeoffs – plant growth was not halted and biosynthesis was improved, especially in trichomes. Is there any (even hypothetical) explanation for this mode of action?
L371: improved SG accumulation
L381: The expression and biosynthesis is improved and growth is not halted. How about molecular mode of action? Where from these differences come from (i.e., why some strains produce such effect while the others do not)? Please try to find if there are any reported species-specific (or strain-specific) effectors of the studied bacteria that hypothetically contributed to the observed diversity of reactions.

·

Basic reporting

Authors have isolated 12 endophytic bacterial strains from different tissues of Stevia and inoculated the plants with these isolates to test the effect of these strains on the plant growth, flavonoid, phenolic and secondary metabolite production. They identified two potential strains which showed higher accumulation of secondary metabolites after inoculation. They also studied the expression of 4 major genes involved in the synthesis of these metabolites.

The authors have provided all the necessary data with sufficient literature background. They have also provided the raw data as supplementary files. The manuscript is well-written but needs a thorough spell check including the table and figure legends. Authors need to include the Table and figure legends with footnotes wherever applicable.

Experimental design

It is understood from the manuscript that authors have isolated the strains from Stevia and re-inoculated the plants with the same strains independently to see the effect of these strains on secondary metabolite production. However, it is not clearly mentioned anywhere in the text so it would be good to describe the methods section with appropriate details.
Methods section should also describe how these 12 different bacterial strains were identified and genotyped.

Additional comments and suggestions are included in the edited version of the manuscript.

Validity of the findings

No comments

Reviewer 3 ·

Basic reporting

In the manuscript, the authors found the endophytic bacteria E. hormaechei H2A3 and E. hormaechei H5A2 increased the SG content and stimulated the density of trichomes in the leaves, as well as the accumulation of secondary metabolites in trichomes. And the accumulation of SG was related to the upregulated of the KO, KHA, UGT74G1 and UGT76G1 genes. There are still many problems with the manuscript. For example, the research objectives are not clear enough, and the results are not discussed in depth. And there are formatting problems in the figure section. So, I suggest that this study be accepted with a major revision. Detailed suggestions are as follows:
INTRODUCTION
1. In this section, it is hard for readers to find what is the main problem to be solved by the study. The key gaps in the literature need to be identified, the key background information presented to highlight these gaps and the way in which this study then proposes to address them.
2. The logical connections between paragraphs need to be reconsidered to allow the reader to better grasp the focus of the study.
3. It would be better to have 2-4 hypotheses here (that arise logically from the introduction). This would also aid the discussion which could then be structured around each hypothesis.

Experimental design

MATERILAS AND METHODS
4. L107, Were these bacteria isolated in this study? If so, please describe the bacteria isolation method and medium. If not, the corresponding references should be noted. In addition, what is the purpose of this research with bacteria from different parts of plants?
5. L113, a reference is lacking here.
6. L114, what is the inoculation method of bacteria? Inoculate in the root? Leaves or soil?

Validity of the findings

RESULTS
7. It is suggested to distinguish ‘Stevioside’ and ‘Rebaudioside A’ in Figure 1 instead of using the same annotation method
8. ‘Bars represent the mean ± standard deviation of six individual plants harvested to 30 days postinoculation.’ This sentence is repeated twice in the text (Figure 1).
9. The Figure note is only half written (Figure 3).
10. ‘Control is plants that were not noninoculated.’ This sentence is repeated twice in the text (Figure 4). And the Alphabetic labels are suggested to be enlarged.
11. The Figure notes of figure5 and 6 also have the problem of sentence repetition. The author should be more carefully in the manuscript written.
12. The font size of alphabetic labels should be unified, because the letter ‘n’ looks smaller than other letters.
13. L273, what is the meaning of ‘if’?
14. L283, a reference is lacking here.

Additional comments

Discussion
15. The discussion would better be structured around your hypothesis, instead of the detected indicators. The main findings should be pointed out for the reader at first and alerting them to the key points that will be discussed.
16. The key, novel, findings need to be emphasized for the reader.
17. L299, I think it's hard to draw a conclusion of ‘growth promotion is not associated with endophytic bacterial’ . This is only the results of the bacteria tested in this study.
18. L324, the colonization related results were not found in the results section.
19. L370, a reference is lacking here.
20. How do the authors explain that H2A3 inoculated plants have higher SG content and density of trichomes, but the expression of SG synthetic genes is lower than that of H5A2?
21. Authors are advised to discuss their results in depth rather than listing similar studies. Pointing out the research significance, emphasizing innovation, and discussing the future research direction.

---

## Round 0.2 · Minor Revisions

Although the reviewers and I thought this paper basically acceptable, the reviewers have some minor corrections to suggest, please refer to the report and attachment for revision. Also, there are some suggested changes to your abstract and some conclusions in the manuscript: reflect the difference between functional and non-functional strains. Here is the section editor's comment for reference:

> The Conclusions
> "Conclusions
> Endophytic bacteria inoculated in S. rebaudiana plants did not promote plant growth, but the bacteria E. hormaechei H2A3 and E. hormaechei H5A2 increased the SG content and stimulated the density of trichomes in the leaves as well as the accumulation of specialized metabolites in trichomes." are confusing and do not agree with the title "Enhanced specialized metabolite ... with endophytic bacteria."
>
> This is very confusing; please explain better the difference between stimulating/non-stimulating endophytes and avoid generalization. In other words, please be specific and clear throughout the manuscript, but especially in conclusions, about the different types/species of endophytes.

Reviewer 1 ·

Basic reporting

The Authors introduced great number of corrections. From my point of view, the Authors adressed all questions very well and supported all the data and analyses that I requested.
The manuscript is very well prepared and references fully support the findings. The Authors prepared new set of figures and tables which are of very high quality.
The hypotheses are now clearly stated and the findings match them.
Overall, the Authors presented outstanding understanding of the topic. I recommed this work as ready for publication with one very minor change that can be introduced during proofing (see comment below).

Experimental design

The hypotheses are clear and concise now. The Authors improved 'Material and Methods' section and supported all the information pertaining to the growth conditions, inoculants and inoculation procedure as well as determination of specialized metabolites. The technical details supported by the Authors strenghted the manuscript in terms of transparence of the methods they used.

Validity of the findings

The Authors introduced new type of statistical anlysis that was requested in my first report. The analyses presented are in-depth and the reader is able to find a full set of information. The conculusions are supported very well by the analyses.

Additional comments

The only change that I request is change of cultivar naming. According to the taxonomical code, name of cultivars should be presented as 'Monaco' and 'Nirmal' but not cv. Monaco or cv. Nirmal. I realize that such nomenclature is used in many works, but it is worth to use correct cultivar naming in such elaborated study.

·

Basic reporting

The authors have addressed all the comments and they are satisfactory. There are few minor corrections that need to be incorporated in the manuscript. The suggestions are included to the edited version of the manuscript.

Experimental design

No comments

Validity of the findings

No comments

---

## Round 0.3 · accepted · Accept

After this revision, I think the conclusion is more accurate. Congratulations on the publication of your article.